# Numerical Investigation on Local Scour and Flow Field around the Bridge Pier under Protection of Perforated Baffle and Ring-Wing Plate

Yan Wang *, Jinchao Chen, Zhihang Wang, Ziqi Zhu and Junxiong Yan

Institute of Foundation and Structure Technologies, Zhejiang Sci-Tech University, Hangzhou 310018, China
* Correspondence: an_wangyan@zstu.edu.cn; Tel.: +86-180-5716-7868

**Abstract:** In this paper, a series of numerical experiments are carried out on the anti-scour device combined with perforated baffle and ring-wing plate. In addition, the optimal dimensions and location of the combined device are obtained: The perforated ratio of the baffle is S = 20%, the distance from the center of the bridge pier is L = 2d (d is the diameter of the bridge pier), and the ring-wing plate is located at H = 1/3h (h is the water depth). To verify the effect of the anti-scour device, the scour characteristics and flow field are further investigated. Compared with single pier and single ring-wing plate, the results revealed that the combined device with the optimal dimensions is of great anti-scour performance. Moreover, the maximum scour depth at the front and side of the pier reduced by 84.20% and 78.95%, which is better than the single ring-wing plate and other combined conditions in the orthogonal experiments. Due to the diversion of perforated baffle and ring-wing plate, the flow velocity at the pier side near the bed surface decreases by 30.7%, and the down-flow is almost eliminated on the vertical plane. Furthermore, the turbulent kinetic energy at different horizontal and vertical planes is reduced due to the reduction in horseshoe vortex and wake flow. Based on the investigation presented herein, the combined device is a promising tool for mitigating scour around the bridge pier.

**Keywords:** combined device; scour hole; velocity vectors; streamlines; turbulence kinetic energy

## 1. Introduction

Local scour around the pier is the main cause of bridge failures [1–5]. Previous studies showed that horseshoe and down-flow are leading concerns for scouring hole creation around the bridge pier. Due to the turbulent fluctuations and accelerated flow, the sediment will be entrained and transported, which will lead to a reduction in the embedment depth and insufficient bearing capacity of the pier. The creation mechanisms of horseshoe and down-flow due to the collision of flow with the bridge pier are presented in Figure 1. In this case, there is a growing body of scholars that recognize the importance of local scour [6–9]. However, there is an urgent need to understand the characteristics of the flow fields around the bridge pier and the scour mechanisms, in parallel, to propose different counter measures to control and reduce local scour around the bridge pier.

A considerable number of literatures on scour protection have been published. These studies show that all the countermeasures can be classified into armoring devices and flow-altering devices. However, existing research showed some disadvantages of single countermeasures. As for the first category, which is usually carried out by the riprap, Beg and Beg concluded that riprap is not an economical method particularly near the bridge site [10]. Moreover, Wang et al. explained that during a flood event, due to the limited improvement in nominal resistance, the riprap is not stable enough to withstand high approaching stream velocities and there is still the possibility of being scoured [11]. Rahimi et al. indicated that debris accumulation upstream of pier will deepen the scour hole [12]. The second way is generally achieved by installing a circular collar or vertical

wall around the pier. Zarrati et al. indicated that wide collars placed at the separate pier may influence the navigation [13].

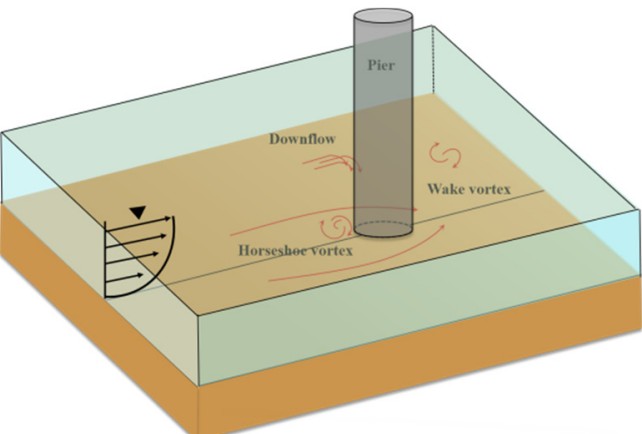

**Figure 1.** Sketch of flow pattern around the pier.

Considering the above studies, some scholars proposed combined countermeasures to improve the anti-scour effect of a single device. Moncada-M et al. combined a slot and a 2b-wide collar placed at the original bed level and achieved a similar 100% reduction in scour depth [14]. Mashahir et al. reached the size and extent of stable riprap around the rectangular pier, which is protected by a collar [15]. Grimaldi et al. tested the best combination of downstream bed-sill and slot, whereby the scour in front of the pier was reduced by 45% [16]. Vijayasree et al. tested the efficiency of combinations of raft foundation with aprons as a scour protection strategy for bridge pier [17]. At a constant flow discharge, Sarkar et al. provided the near-bed turbulence statistics over the deformed bed generated around the submerged cylindrical pier embedded vertically on a loose sediment bed [18].

In parallel, a comprehensive understanding of the flow field is necessary to accurately predict the scour magnitude. Ataie-Ashtiani and Aslani-Kordkandi investigated the structure of the local flow field around the side-by-side bridge pier with and without scour hole [19]. Keshavarzi et al. conducted an experimental study of the turbulent flow interaction around two in-line circular piers with varied spacing [20]. In the past few decades, although there has been a large number of literatures on the scour around the cylindrical pier, significant challenges still exist in regards to the flow field around the pier, which is under the protection of combined device. However, the continuous advances in computer technology and finite element simulation allow for a description of the complex flow field to be obtained in recent years [21,22].

In this paper, on the basis of the ring-wing plate proposed by Cheng et al. [23] and considering the reduction in the velocity at two upstream corners of a pier, a device combined of perforated baffle and ring-wing plate is proposed. A series of comprehensive numerical tests have been performed using the orthogonal method to find the optimal dimensions and location of the device for achieving the maximum reduction in scour depth. Additionally, the flow field around the protected pier is further investigated to better understand the anti-scour mechanisms. All the simulation results are compared with those of single pier and only under the protection of ring-wing plate. It is expected that these new observations on the flow structure will be useful for understanding the mechanisms of local scour and designing corresponding protective devices in ocean engineering.

## 2. Material and Methods

### 2.1. Definition of Variables and Test Planes

As presented in Figure 2, the experimental setups in this paper include a circle pier and combined device. As for the combined device, the front end of the perforated baffle is located at a distance of L from the center of the pier and the ratio of perforated area is S.

The ring-wing plate is at H, which is fixed on the pier. Of note, Huang et al. obtained the optimal parameters of the surface guide panels, which are the interior angle of $\theta = 60°$ with $L/D = 2 - 2.5$ [24]. Similarly, Wang et al. retrofitted $60°$ of V-shaped baffle with a wing plate to enhance the anti-scour effect on a circle pier [25]. In view of the above experience, the angle of the perforated baffle in this paper is determined to be $60°$, namely, the angle is not regarded as an optimization parameter. Based on the outcomes of Cheng et al. [23], the extension length of the ring-wing plate is consistent with the radius of the pier, and three values of H are considered for 1/2, 1/3, 1/6h.

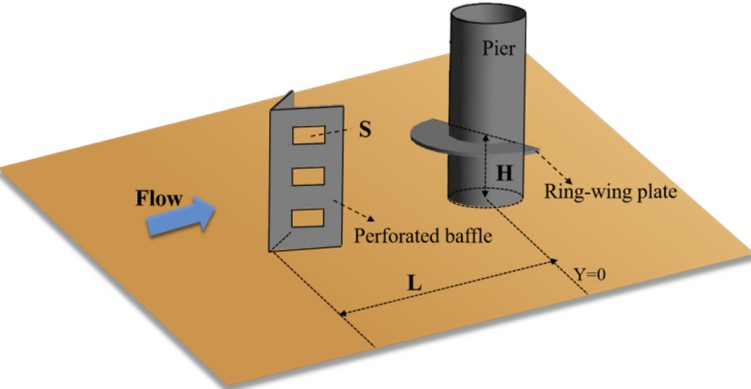

**Figure 2.** Sketch of the experimental setup.

As a statistical methodology, the orthogonal experiment has the advantages of covering a wide range of variables to be tested with a minimum number of tests. The experimental variables with their values and levels are listed in Table 1. The single pier and single ring-wing plate are set as the control group, the rest are set up in 9 groups according to the rules of the orthogonal experiment. In addition, the combined device with optimal dimensions, which is derived from the orthogonal experiment, is set as the validation group, run no. 12 in Table 2. The parameters and results of each group are summarized in Table 2.

**Table 1.** Experimental variables with values and levels.

| Levels | Factors | | |
|---|---|---|---|
| | S | L | H |
| 1 | 10% | 2d | 1/6h |
| 2 | 20% | 3d | 1/3h |
| 3 | 30% | 4d | 1/2h |

**Table 2.** Experimental design and the experimental results.

| Protection Level | Run No. | Factors | | | Scour Depth (cm) | |
|---|---|---|---|---|---|---|
| | | S | L | H | Front End of Pier | Side End of Pier |
| I | 1 | - | - | - | 3.16 | 1.9 |
| II | 2 | - | - | 1/3h | 1.2 | 0.6 |
| | 3 | 10% | 2d | 1/2h | 0.8 | 0.9 |
| | 4 | 10% | 3d | 1/3h | 0.9 | 0.7 |
| | 5 | 10% | 4d | 1/6h | 1.1 | 0.8 |
| | 6 | 20% | 2d | 1/6h | 0.8 | 0.8 |
| | 7 | 20% | 3d | 1/2h | 0.6 | 0.7 |
| III | 8 | 20% | 4d | 1/3h | 0.6 | 0.8 |
| | 9 | 30% | 2d | 1/3h | 0.7 | 0.6 |
| | 10 | 30% | 3d | 1/6h | 0.9 | 0.9 |
| | 11 | 30% | 4d | 1/2h | 0.9 | 0.6 |
| | 12 | 20% | 2d | 1/3h | 0.5 | 0.4 |

As indicated in Figure 3, the interface of water and sand is set as Z = 0. In addition, to observe the flow characteristics clearly, there are two horizontal (Z = 1 cm and Z = 5 cm) and two vertical (Y = 0 cm and Y = −2 cm) planes set in the flow field.

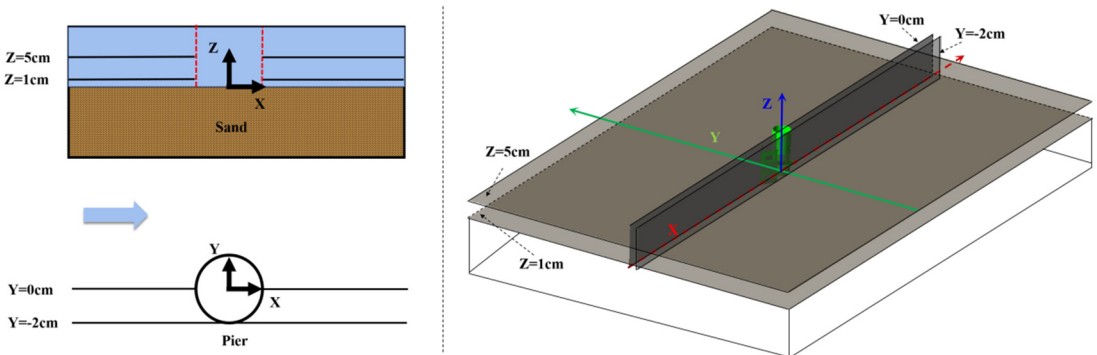

**Figure 3.** Test planes of flow field.

*2.2. Computational Setup*

2.2.1. Governing Equations

The fluid equations for unsteady, incompressible, three-dimensional viscous water flow around bridge piers can be solved approximately by the Reynolds-averaged Navier-Stokes equations. In addition, the standard k-ε turbulent model will be a desired selection for riverbed- and pier-surface-bounded three-dimensional unsteady flow.

Here, k and ε can be obtained from the following transport equations:

$$\rho \frac{\partial k}{\partial t} + \rho u_i \frac{\partial k}{\partial x_i} = \frac{\partial}{\partial x_i}\left[\left(\mu + \frac{\mu_t}{\sigma_k}\right)\frac{\partial k}{\partial x_i}\right] + G_k - \rho\varepsilon \tag{1}$$

$$\rho \frac{\partial \varepsilon}{\partial t} + \rho u_i \frac{\partial \varepsilon}{\partial x_i} = \frac{\partial}{\partial x_j}\left[\left(\mu + \frac{\mu_t}{\sigma_k}\right)\frac{\partial \varepsilon}{\partial x_j}\right] + C_{1\varepsilon}\frac{\varepsilon}{k}G_k - \rho C_{2\varepsilon}\frac{\varepsilon^2}{k} \tag{2}$$

where $G_k = \mu_t(\frac{\partial u_i}{\partial x_j} + \frac{u_i}{x_j})\frac{\partial u_i}{\partial x_j}$ is the source term of turbulent kinetic energy. $C_{1\varepsilon}$, $C_{2\varepsilon}$, $C_\mu$, $\sigma_k$, $\sigma_\varepsilon$ are empirical constants, with values of $C_{1\varepsilon} = 1.44$, $C_{2\varepsilon} = 1.92$, $C_\mu = 0.09$, $\sigma_k = 1.0$, and $\sigma_\varepsilon = 1.3$, respectively.

A second-order upwind scheme is employed for the convective term, while the viscosity term uses a second-order central difference scheme. The SIMPLEC algorithm is employed to solve the discretized equations.

2.2.2. Description of Computational Domain

Referring to the flume test [26], the 3D model in this paper has a total length of 1.2 m, a width of 0.8 m, and a height of 0.3 m. Melville and Coleman indicated that the flume width should be at least 10 times greater than the pier diameter to minimize any contraction effects on the scour depth [27]. In this study, the flume width is 0.8 and the 0.04 m pier diameter was selected. Therefore, the resultant ratio of the flume width to the pier diameter was 20 and it satisfied this criterion. The center of the pier on the interface is defined as the coordinate origin. The first layer of the model was water (h = 0.1 m), and the sand in the lower layer is 0.15 m, with the grain size of $d_{50}$ = 0.8 mm and density of $2.600 \times 10^3$ kg/m$^3$.

The boundary conditions of the computational domain are as follows: The velocity inlet (v = 0.25 m/s) and outflow are 0.6 m away from the pier. The walls on both sides are 0.4 m away from the pier, except for the three components of velocity, all quantities adopt the zero-gradient condition. The top boundary is defined as symmetry. In addition, the perforated baffle, ring-wing plate, and pier are defined as "wall". To achieve the hydraulically smooth surface, the roughness of the "wall" is maintained as sufficiently small ($k_s = 1 \times 10^{-5}$ m).

The present numerical models are 3D cases. The computational domain is discretized into finite volumes of quadrilateral blocks in varying shapes and dimensions. The need to keep the computational time affordable while optimizing the computational mesh for convergence is emphasized. As shown in Figure 4, body-fitted quadrilateral grids were used and refined near the pile and baffle. The total number of cells comprising the utilized computational domain is $2.5 \times 10^5$.

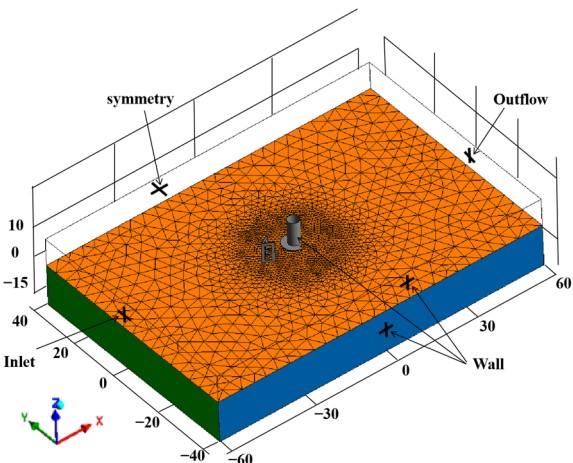

**Figure 4.** Mesh in computational domain and boundary conditions.

## 3. Data Analysis

### 3.1. Numerical Validations

The numerical results are listed in Table 2. To verify the reliability of the numerical simulation and determine the error limits between the data obtained from FLUENT with the physical experimental results, Equation (3) is exploited:

$$e = ((d_s/D)_N - (d_s/D)_E)/(d_s/D)_N \times 100 \tag{3}$$

where e is the relative error, reflecting the consistency between the numerical model and the flume experiment. $(d_s/D)_N$ is the dimensionless scour depth in numerical solution and $(d_s/D)_E$ is the dimensionless scour depth in hydraulic laboratory. Taking the scour depth at the front end of single pier as an example, the physical experiment corresponding to this paper is 3.1 cm [26] and e = 1.899. There is a significantly good accordance between the numerical and physical experiment results.

Figure 5 shows the comparison of simulated and experimental results of streamlines near the riverbed. Compared with Melville and Raudkivi flume test results [28], it can be seen that the position of the simulated streamlines around the front side of the pier and the separation point on the back are similar to the experiment result. Specifically, there is a similar backflow area in the wake of the pier.

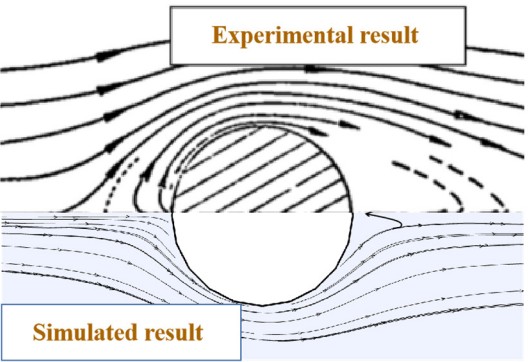

**Figure 5.** Comparison of simulated and experimental results of streamlines.

Therefore, it is reliable to simulate pier scouring and flow field by this model.

### 3.2. Range Analysis of the Orthogonal Experiment Results

Based on Table 2, the results of range analysis for the maximum scour depth are listed in Table 3, where $K_1$, $K_2$, and $K_3$ are defined as the average scour depth of three levels in each factor. R value is the difference between the maximum and minimum value of $K_1$, $K_2$, and $K_3$ for a certain factor, namely, the R for each factor can be obtained according to the following equation:

$$R = K_{max} - K_{min} \tag{4}$$

**Table 3.** Results of range analysis.

| Factors | Range Analysis | | | | | |
|---|---|---|---|---|---|---|
| | $K_1$ | $K_2$ | $K_3$ | R | Optimal Level | Order |
| S | 0.933 | 0.667 | 0.833 | 0.266 | $S_2$ | 1 |
| L | 0.767 | 0.800 | 0.867 | 0.100 | $L_1$ | 3 |
| H | 0.767 | 0.733 | 0.933 | 0.200 | $H_2$ | 2 |

Furthermore, the main-effect plots are illustrated in Figure 6. It can be seen that the variable ranked first is S, and the sequence of the impact of all other variables on the maximum scour depth reduction can be listed as H > L.

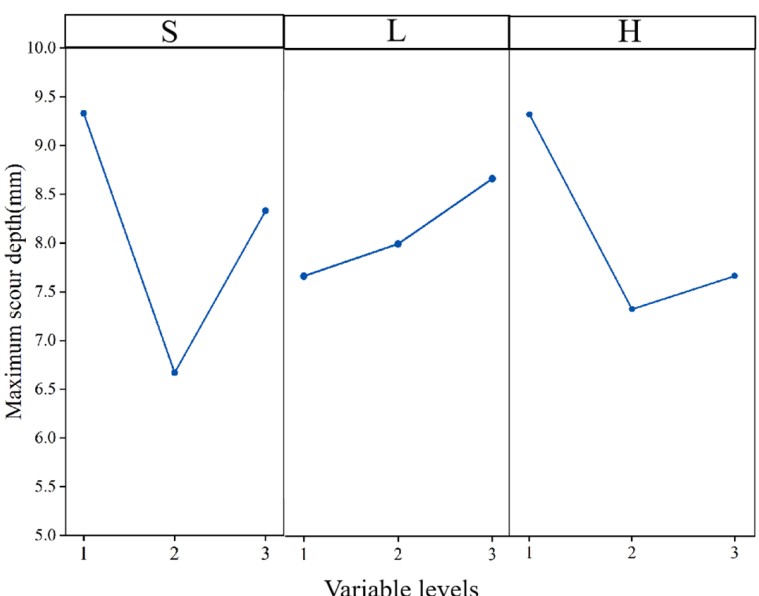

**Figure 6.** Main-effect plots of the scour depth.

For the main aim of this study, the best value for each variable should be determined where the scour depth is the lowest, referring to Table 3 and Figure 6, the optimal ratio of perforated area is 20%. Increasing S may reduce the effectiveness of this structure. However, there is still a question regarding the efficiency when S is lower than 20%. The trend of distance between the pier and the perforated baffle showed that the maximum reduction in the scour depth was achieved when the distance was lowest. Of note, the best value for the height of ring-wing plate is 1/3h, which is consistent with the findings of Cheng et al. [23].

### 3.3. Variance Analysis of the Orthogonal Experiment Results

Although the optimum dimensions and location of the device could be easily obtained by the range analysis, it is necessary to introduce the variance analysis to provide a measure of confidence and further investigate the significance of different factors. Degrees of

freedom (f), sums of squares (SS), mean square (MS), and the variance ratio (F) are analyzed and organized in Table 4. In addition, the F value is used to show the significance of each factor effect.

**Table 4.** Analysis of variance for the maximum scour depth.

| Source | f | SS | MS | F |
|---|---|---|---|---|
| S | 2 | 0.109 | 0.054 | 7.000 |
| L | 2 | 0.016 | 0.008 | 1.000 |
| H | 2 | 0.069 | 0.034 | 4.429 |
| Residual error | 2 | 0.016 | 0.008 | |
| Total | 8 | 0.209 | | $F_{0.05}$ (2,8) = 4.46 |

The results show that the F value of each factor follows the sequence: $F_S > F_H > F_L$, which is consistent with the result of range analysis. Of note, $F_S > F_{0.05}$ and $F_H$ are very close to $F_{0.05}$, indicating that the effects of factor S on the reduction in scour depth is prominent, and the height (H) should also be considered. Therefore, the optimal dimensions and location of the device are obtained as $S_2L_1H_2$: S = 20%, L = 2d, H = 1/3h.

## 4. Discussion

### 4.1. Scour Hole Depth

To further illustrate the anti-scour effect of combined device, the scour depth and reduction rate for all the conditions are shown in Figure 7. As can be seen, under the protection of the combined device, scour depth at the front and side end of the pier are reduced significantly at the same time. In contrast, the values at the front and side of the pier are closer, which shows that the riverbed around the pier is well protected together. The average scour hole is less likely to cause bridge safety accidents. Of note, in the results of run no. 12, the maximum scour depth at the front and side end of the pier reduced by 84.20% and 78.95%, respectively, which is better than the single ring-wing plate and other combined conditions in the orthogonal experiment. This not only shows the reliability of the orthogonal experiment, but also shows the excellent anti-scour performance of this device.

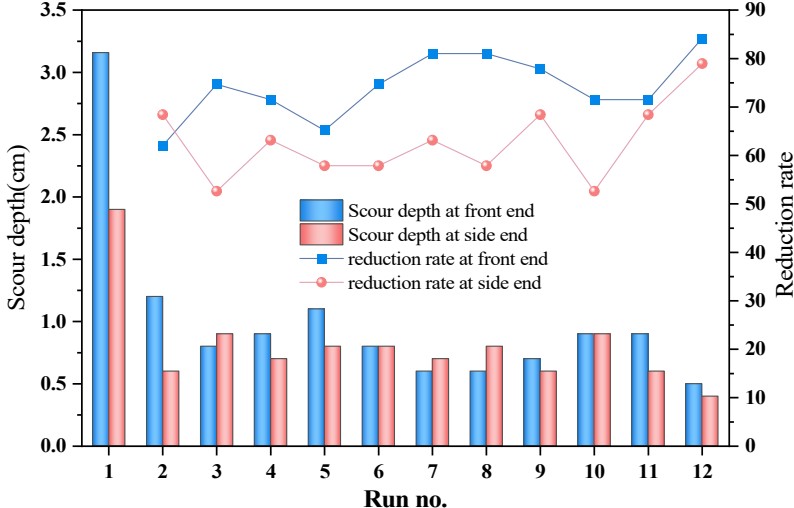

**Figure 7.** Scour depth and reduction rate.

### 4.2. Flow Field at Horizontal Planes

Based on the reliable simulated results, velocity vectors, streamlines, and turbulent kinetic energy will be described and discussed in detail to understand how the optimized combined device affects the flow field and consequently reduces the scour. (In

the following, (a), (b), and (c) represent single pier, single ring-wing plate, and combined device, respectively).

### 4.2.1. Velocity Vectors and Streamlines

In this section, the numerical results for three conditions are presented. Figures 8 and 9 show the velocity vectors at Z = 1 cm and Z = 5 cm. As can be seen, the magnitude and direction of the velocity vectors are $(u^2 + v^2)^{0.5}$ and arctan $(v/u)$, respectively. Since the flow field changes mainly occur around the bridge pier and protective devices, only the local test results are shown.

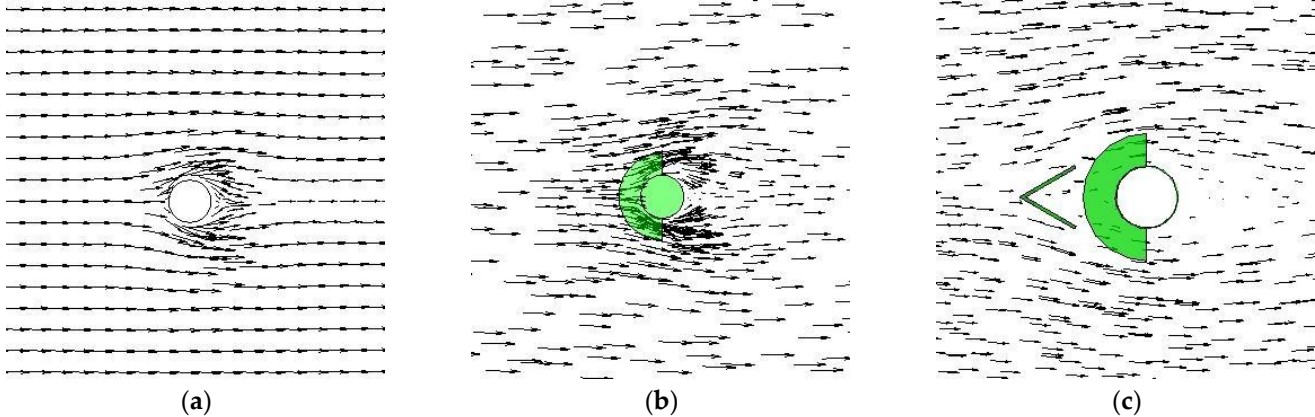

(**a**)         (**b**)         (**c**)

**Figure 8.** Velocity vectors at Z = 1 cm. (**a**) Single pier; (**b**) Pier with ring-wing plate; (**c**) Pier with combined device.

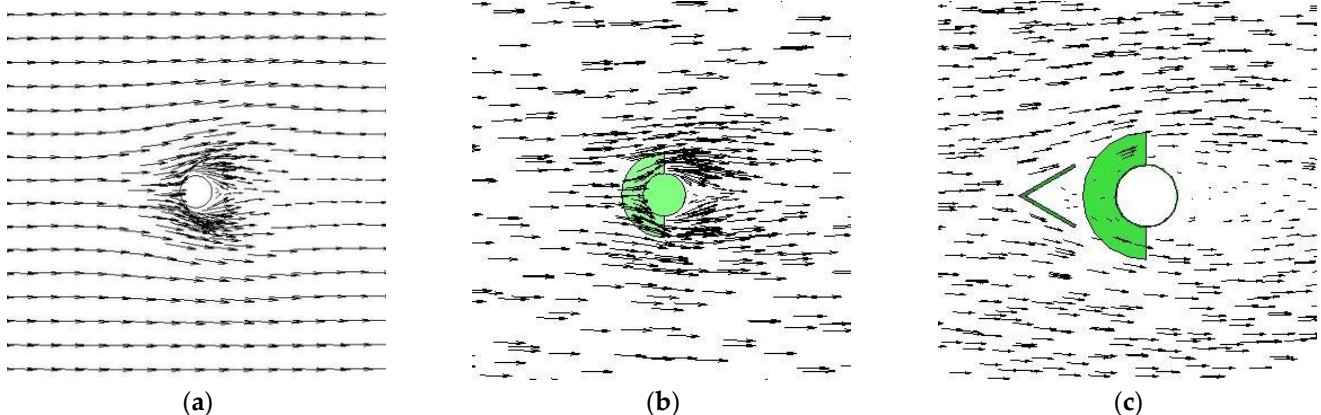

(**a**)         (**b**)         (**c**)

**Figure 9.** Velocity vectors at Z = 5 cm. (**a**) Single pier; (**b**) Pier with ring-wing plate; (**c**) Pier with combined device.

As can be seen in Figures 8a and 9a, due to the adverse pressure gradient induced by the pier, a boundary layer upstream of the pier separates. At both sides of the pier, the separation points of boundary layer shift further to downstream on the upper surface of riverbed. This is consistent with the conclusions of single pier [29–31]. In addition, the turbulence increase near the bed is due to the effect of bed roughness. Due to the turbulent boundary layer resistance on the sides of the pier, the separation to downstream face of the pier is delayed. Therefore, compared with the upper level, this phenomenon results in a smaller wake.

Figures 8b and 9b show the velocity vectors at Z = 1 cm and Z = 5 cm for the bridge pier under the protection of the ring-wing plate. Similar to the single pier, the flow is blocked by the pier and produces a basically symmetrical flow around both sides. At Z = 1 cm, owing to the ring-wing plate blocking the down-flow and the resistance of the rough riverbed,

part of the flow near the bed moves in the direction of separation from the pier, and part of the flow spreads downstream and forms a small-scale vortex. At Z = 5 cm, the ring-wing plate can be regarded as a relatively smooth and a non-erodible surface, the horseshoe vortex generated by the down-flow colliding the baffle meets part of the incoming flow behind the pier. In parallel, part of the flow separated at both sides of the pier and travels along the incoming flow direction. However, it is undeniable that the existing horseshoe and wake vortices, together with the influence in gravity and bed friction, will increase the likelihood of the sediment deposition downstream of the pile.

Under the protection of the combined device, the velocity vectors around the pier are presented in Figures 8c and 9c. Of note, the perforated baffle upstream of the pier is able to weaken the horseshoe vortex by diverting a part of the near-bed flow and decreasing the velocity of incoming flow. As can be seen, the flow around the pier and the vortex downstream of the pier are greatly reduced, which induces lower bed shear stress. When the bed shear stress is less than the critical shear stress, the material on the bed surface may not be scoured.

The velocity contours and the streamlines around the pier under three conditions are shown in Figures 10 and 11. In general, the distributions are reasonably symmetrical at the left and right sides of the pier. In addition, there is a separated flow at the front end of the pier and baffle, which is the main factor that induces reverse-vortex flows downstream of the pier. Moreover, it can be seen that the velocity near the bed is lower than the middle layer of the flow, this situation can be attributed to the existence of the horseshoe vortex and the roughness of the sediment surface. However, the flow field under the protection of the ring-wing plate is different, more specifically, there are still areas of high velocity on both sides of the pier at these two planes.

As for the single pier, the streamlines around the pier are similar while the velocity is different at these two planes. The sediment particles began to transfer to the downstream as the flow collides the pier and the horseshoe vortices, which decreases the velocity at Z = 1 cm. In addition, due to the blocking of the pier, the velocity in front of and behind the pier decreases. Moreover, it can be seen that a symmetrical acceleration region is formed on both sides of the pier. The length of this region in the Y-axis is about 1.5d, and the velocity is greater than the inlet velocity (v = 0.25 m/s), especially at Z = 5 cm, the velocity at both sides of the pier is about 0.34 m/s. It can be speculated that this will increase the generation of horseshoe vortices and the possibility of sediment transportation. After installing the ring-wing plate, the symmetrical flow still exists, which is presented in Figures 10a and 11b. The streamlines and velocity contour near the bed are similar to the single pier. However, the region with high velocity shrinks at Z = 5 cm, which is due to the fact that the ring-wing plate divides the incoming flow into two parts in front of the pier, and only the flow above the plate turns into the accelerated flow around the bridge. As can be seen in Figures 10c and 11c, under the protection of the combined device, a region of slower velocity (0.02~0.04 m/s) is formed between the perforated baffle and the pier. Of note, the area is larger than the first two conditions. It can be concluded that the presence of diverted flow upstream of the perforated baffle weakens the contraction of the streamlines induced by the horseshoe vortex and reduces the influence of velocity amplification. Based on this fact, it can be asserted that the likelihood of sediment to be swept up by the flow will reduce. Moreover, it can be seen that two "leaf-shaped" regions with high velocity are created near the intersection of baffle extension direction and Y-axis (X = 0) at the horizontal plane. Particularly near the bed surface, the velocity in this region is not only lower than run no. 2, but also lower than the velocity of the inlet. Of note, this region is not connected to the pier and there is a separation between the two. This shows that the perforated baffle has the effect of protecting the riverbed at both sides of the pier. In addition, the velocity behind the pier is greatly reduced, and the area is larger, about 2d long in the X-axis. As for the streamlines, compared with the first two conditions, the location of the convergence of the winding flow to the back of the pier is relatively distant, thus it reduces the vortex behind the pier and the low-velocity region behind the pier is

also relatively larger. This finding, together with other conclusions in the present study, strongly show that the combined device protects the sediment around the pier effectively.

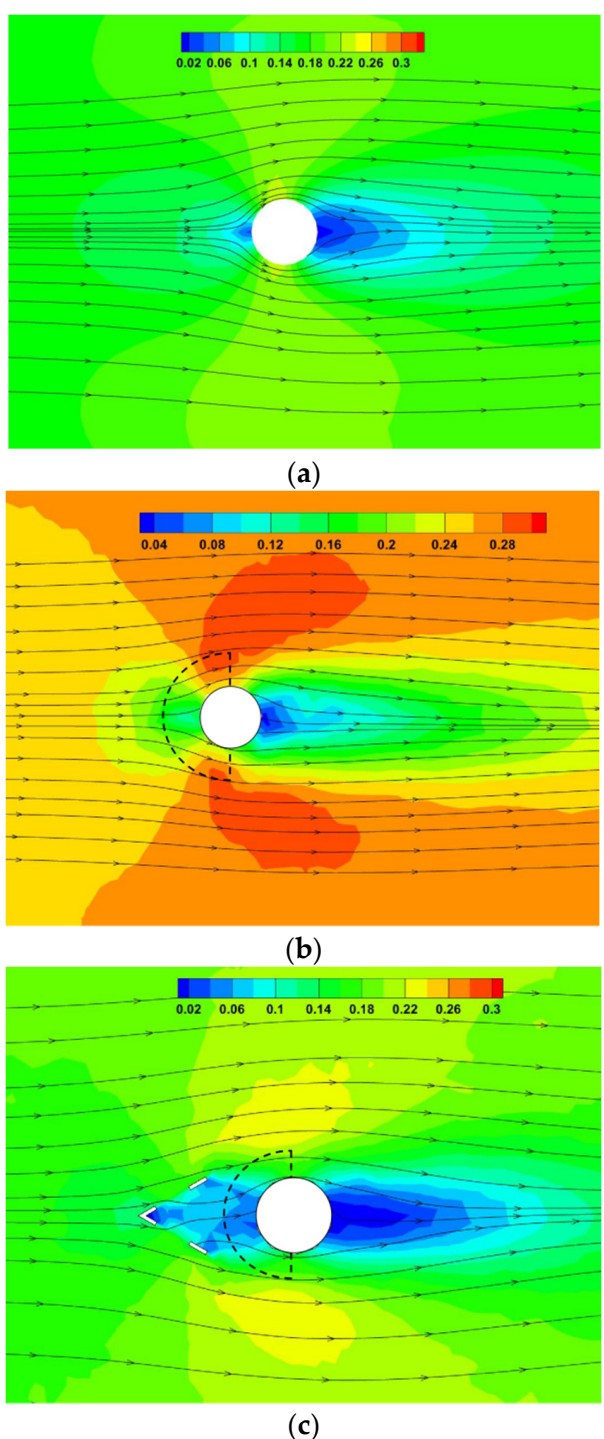

**Figure 10.** Streamlines and velocity contours at Z = 1 cm. (**a**) Single pier; (**b**) Pier with ring-wing plate; (**c**) Pier with combined device.

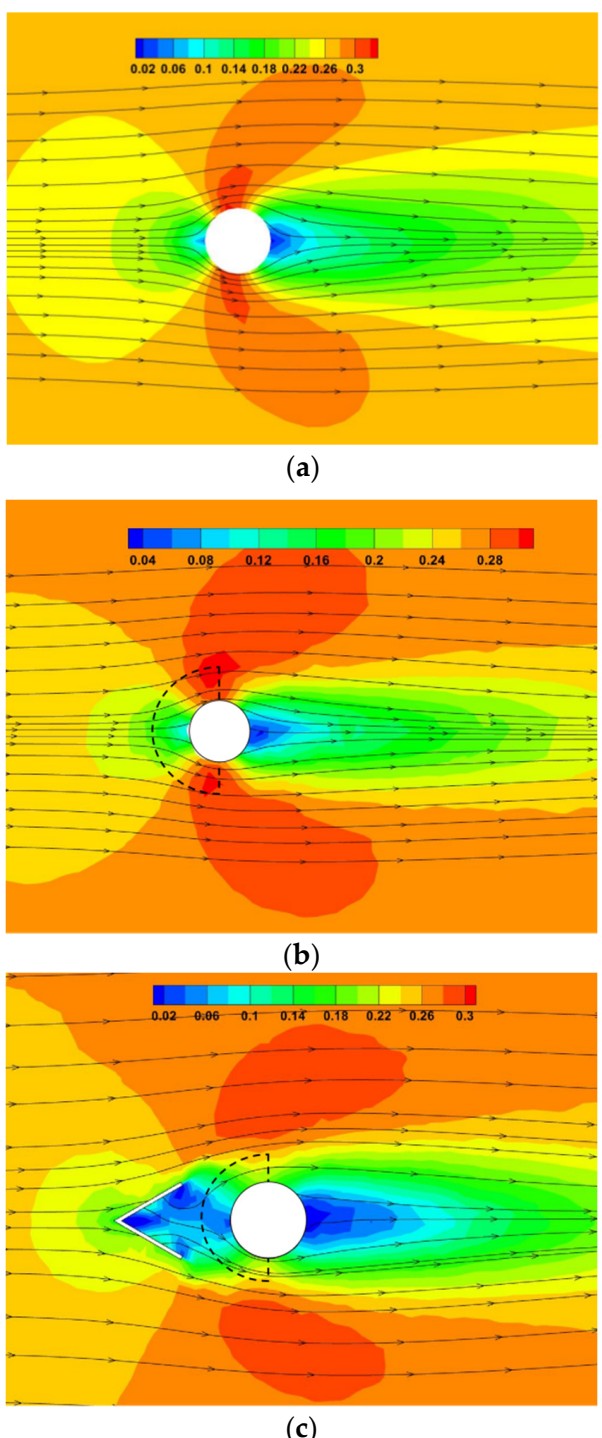

**Figure 11.** Streamlines and velocity contours at Z = 5 cm. (**a**) Single pier; (**b**) Pier with ring-wing plate; (**c**) Pier with combined device.

### 4.2.2. Turbulence Kinetic Energy

Figures 12 and 13 show the turbulent kinetic energy at Z = 1 cm and Z = 5 cm, respectively. On the whole, the turbulent kinetic energy near the bed (Z = 1 cm) is significantly greater than Z = 5 cm. As shown in Figure 13a, there is a small turbulent kinetic energy within 8d behind the pier, and the region is oval. After the installation of the ring-wing plate, a region of high turbulent kinetic energy appears in front of the pier at these two planes. In addition, at the sides of the pier near the bed, a discontinuous band of high turbulence region with 60° to the incoming flow direction appears, which is shown in the

dashed line. Of note, a very important conclusion is that the presence of the ring-wing plate makes the flow around the pier interact more strongly with the incoming flow, creating more vortices in the boundary separation direction. As indicated in Figures 12c and 13c, the turbulent kinetic energy around the pier at both planes is greatly decreased. Indeed, the low turbulent kinetic energy region presents a circle with a radius of about 20 cm. At the plane of Z = 1 cm, there is a low turbulent kinetic energy region with a length of about 4d behind the pier. Of note, the turbulence kinetic energy behind the baffle is about 0, which is a rational consequence of the blocking effect of perforated baffle and pier.

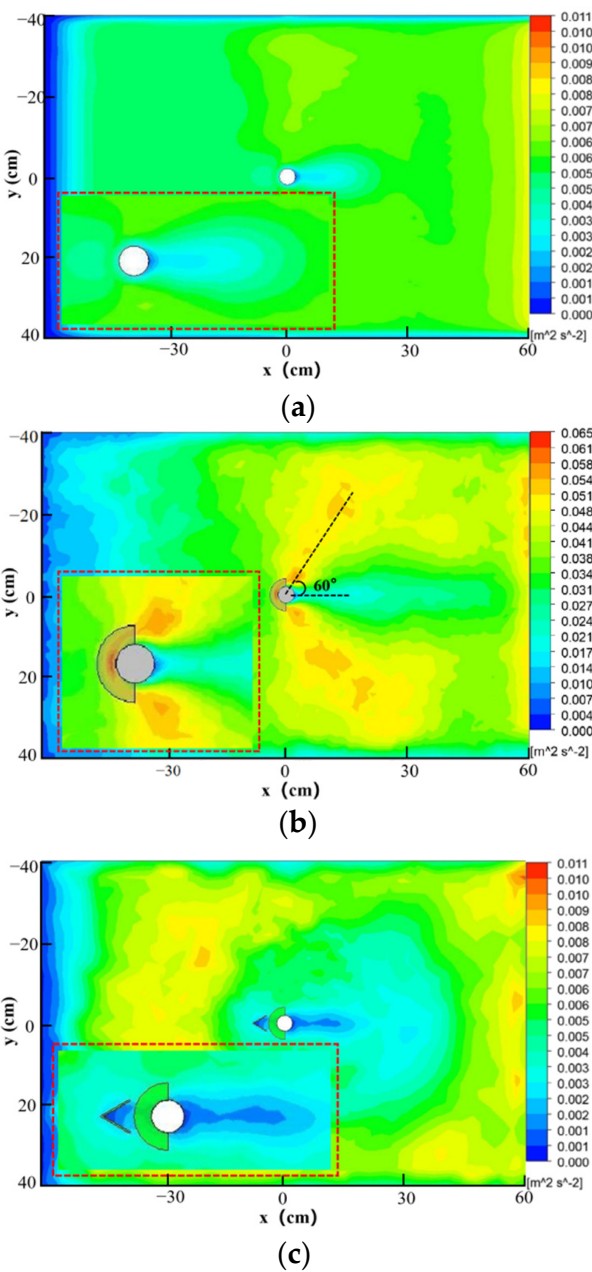

**Figure 12.** Turbulence kinetic energy at Z = 1 cm. (**a**) Single pier; (**b**) Pier with ring-wing plate; (**c**) Pier with combined device.

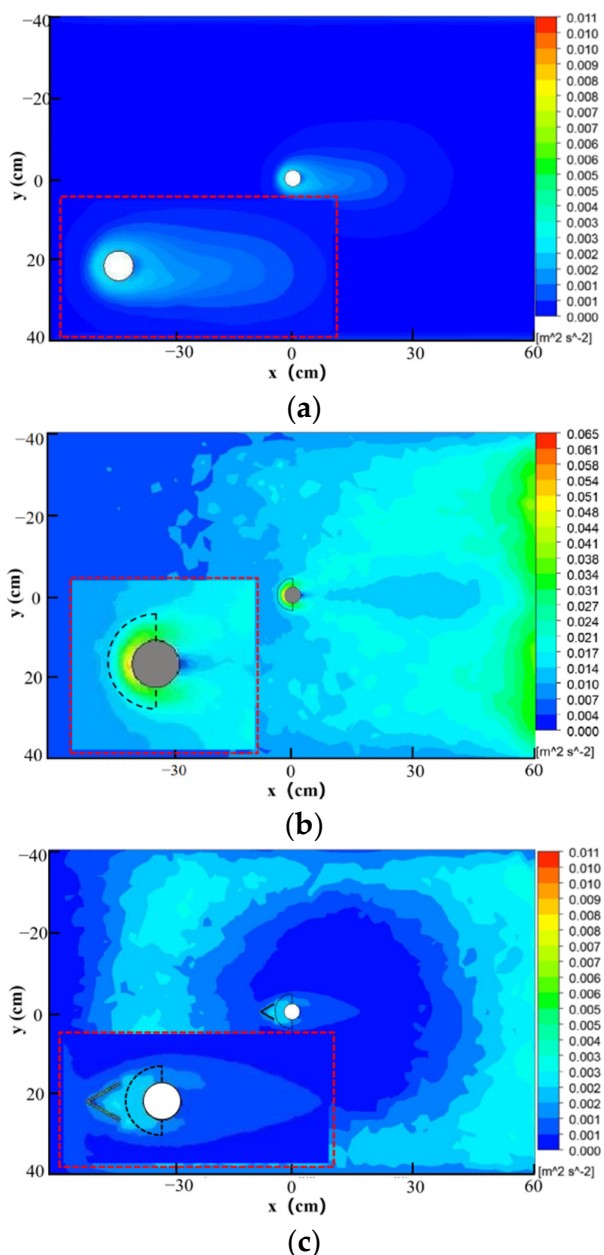

**Figure 13.** Turbulence kinetic energy at Z = 5 cm. (**a**) Single pier; (**b**) Pier with ring-wing plate; (**c**) Pier with combined device.

*4.3. Flow Field at Vertical Planes*

4.3.1. Velocity Vectors and Streamlines

Figures 14 and 15 show the velocity vectors around the pier at the XZ plane (Y = 0) under three conditions. Figures 16 and 17 show the streamlines around the pier with the vertical velocity contour in the background. As can be seen in Figures 14a and 15a, under the free surface, the velocity decreases and consequently the pressure increases as the flow approaches the pier. Moreover, the increase in pressure decreases in a vertical downwards direction. Owing to the approaching boundary layer flow, an adverse pressure gradient on the upstream face of the pier in the vertical direction is formed by this difference in the pressure. Consequently, a down-flow on the upstream face of the pier is generated. On the side, the trend of down-flow in front of the pier is weakened, while behind the pier, the flow shows an upward trend of acceleration.

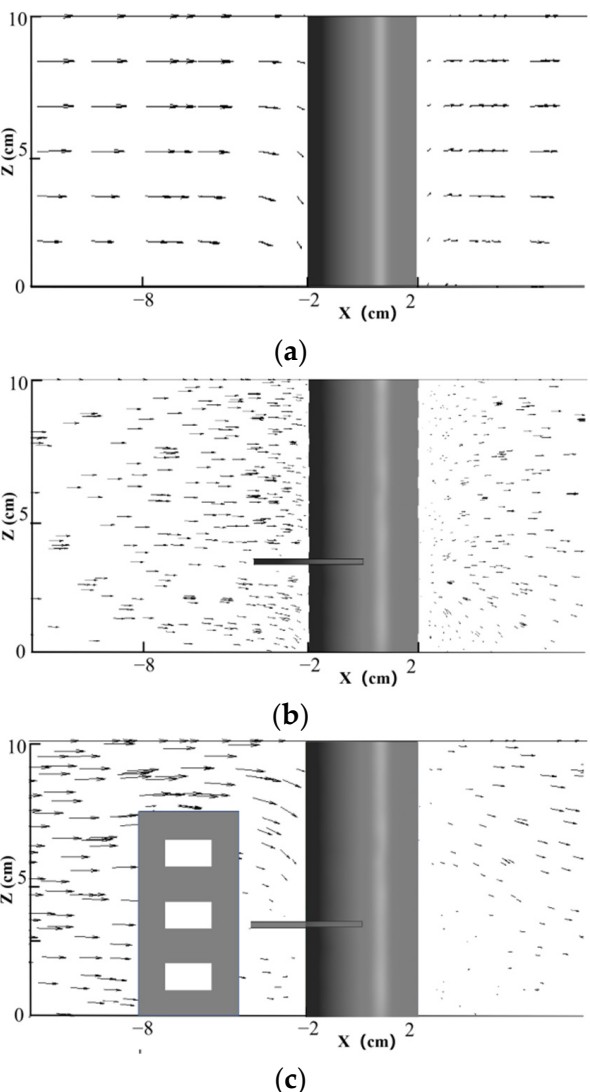

**Figure 14.** Velocity vectors at Y = 0. (**a**) Single pier; (**b**) Pier with ring-wing plate; (**c**) Pier with combined device.

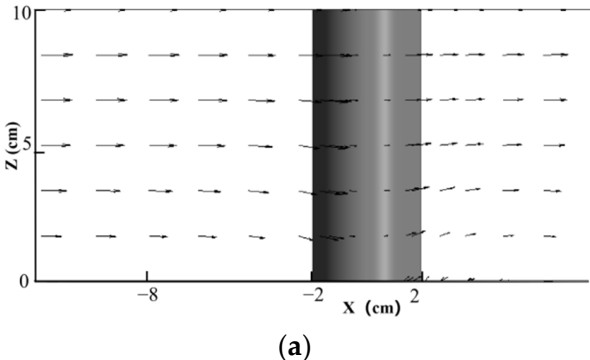

**Figure 15.** *Cont.*

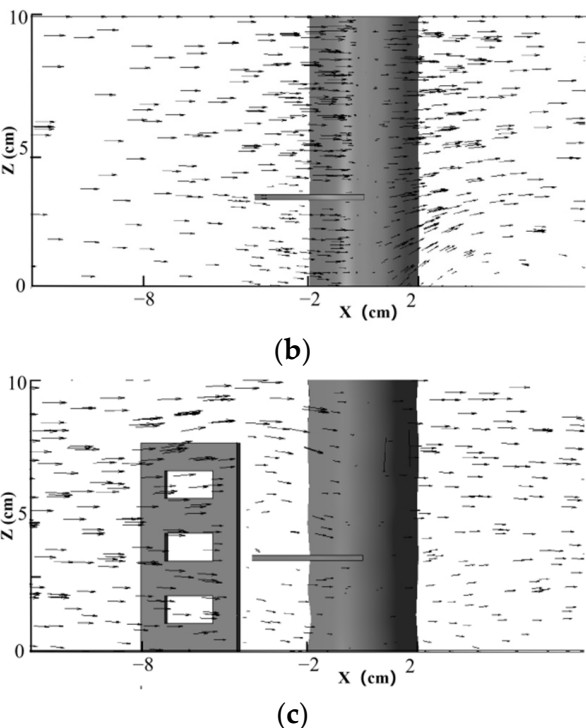

**Figure 15.** Velocity vectors at Y = −2 cm. (**a**) Single pier; (**b**) Pier with ring-wing plate; (**c**) Pier with combined device.

As can be seen in Figures 16b and 17b, due to the blocking effect of the ring-wing plate, the region with high velocity under the plate shrinks. Moreover, it can be seen that the ring-wing plate has a certain protective effect to the sediment near the riverbed in front of the pier. However, compared with the single pier, the incoming flow has a downward trend earlier and begins to decline directly below the front end of the plate. At the rear of pier, there is an important difference in the velocity patterns between the single pier and ring-wing plate pier. As indicated in Figure 16b, a negative velocity region appears at the rear of pier, namely, the riverbed behind the pier will be scoured. At the vertical plane, although the area of region with high velocity is reduced, the velocity near the riverbed behind the pier is still greater than the inlet velocity, which may also lead to the possibility that the sediment behind the pier may be taken away. Therefore, a single ring-wing plate cannot prevent scouring completely.

As expected, after installing the perforated baffle, the flow structure dramatically changes upstream of the pier, which is shown in Figures 16c and 17c. Another significant outcome that can be drawn here is that there is an upward flow at the upper end of the perforated baffle. More specifically, a wake region forms behind the perforated baffle. These wake vortices and upward flow prevent the down-flow from impinging on the riverbed upstream of the pier. In addition, the horseshoe vortices are disturbed by this wake. Therefore, the reduction in scour depth can be attributed to these changes in the flow field around the pier.

In addition, as can be seen, the existence of the perforated baffle in the combined device makes the negative velocity region under the ring-wing plate disappear. In particular, the flow under the plate is basically parallel to the incoming flow. Although there is still a down-flow behind the pier, its velocity is very small (−0.02 m/s). On the side, the phenomenon of acceleration behind the pier also disappears, namely, the streamlines are basically parallel to the incoming flow. The findings above reveal that the combined device has an excellent anti-scour effect on the riverbed around the pier.

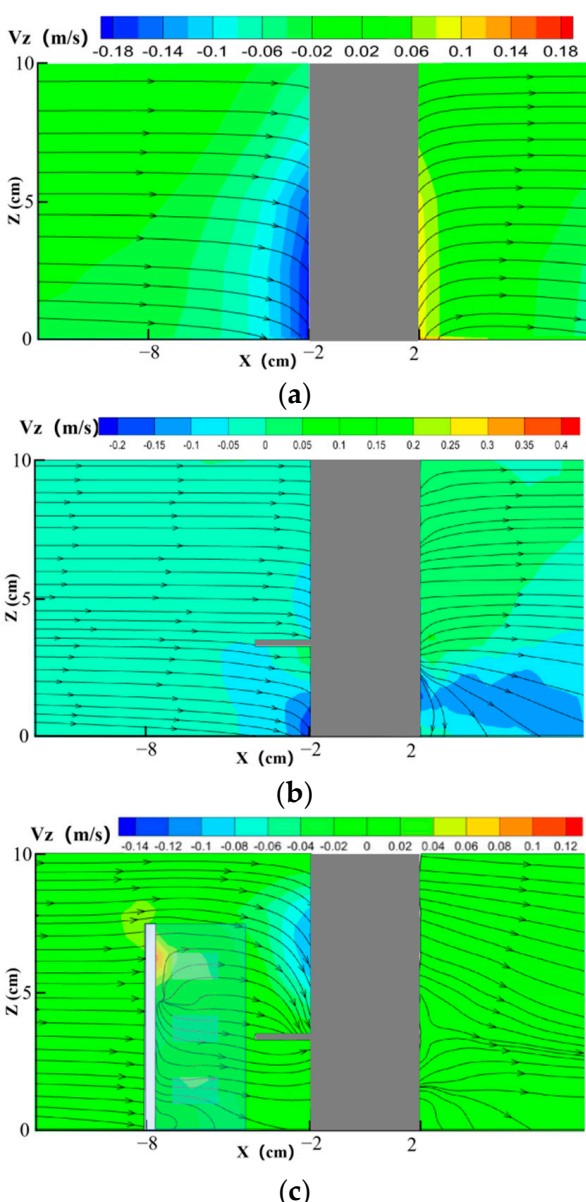

**Figure 16.** Streamlines and velocity contours at Y = 0.

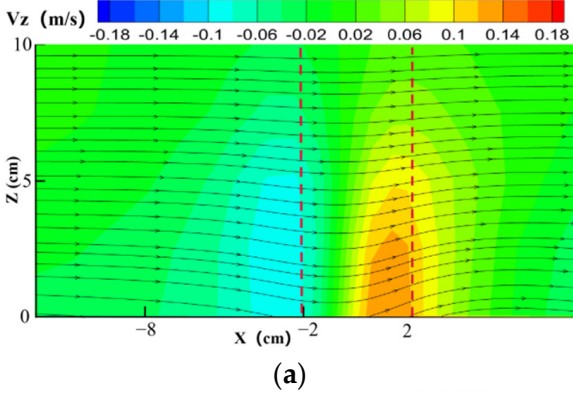

(**a**)

**Figure 17.** *Cont.*

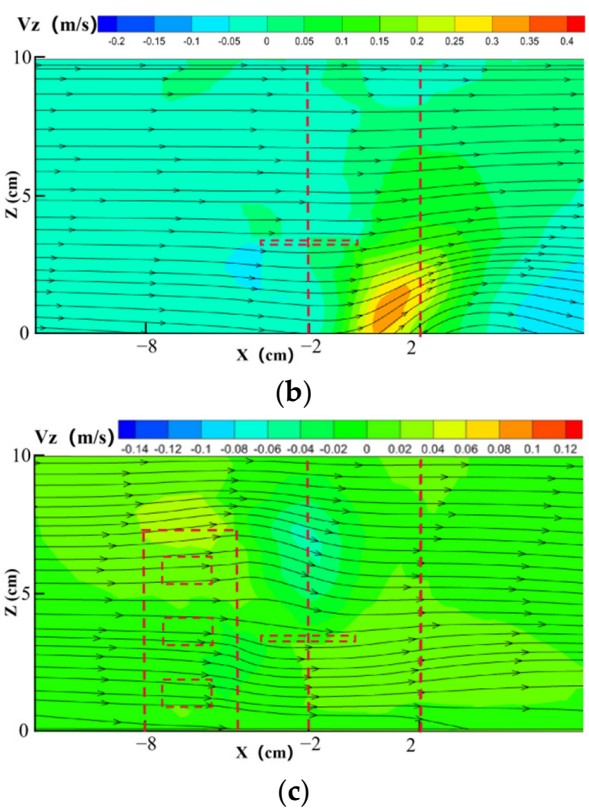

**Figure 17.** Streamlines and velocity contours at Y = −2 cm. (**a**) Single pier; (**b**) Pier with ring-wing plate; (**c**) Pier with combined device.

### 4.3.2. Turbulence Kinetic Energy

Figures 18 and 19 show the turbulent kinetic energy at vertical planes. In an overall assessment, the turbulent kinetic energy under the protection of the combined device is significantly smaller than the first two conditions, which can be seen from the color of the contour. As for the single pier, there is a high turbulent kinetic energy region in front of the pier and near the bed. That is due to the fact that the velocity decreases and consequently the pressure increases as the flow approaches the pier, which leads to the down-flow on the upstream face of the pier. In addition, the down-flow interacts with the riverbed and rolls up afterwards to create a horseshoe vortex. Moreover, it may be formed by an interaction with the surrounding flow at both sides of the pier.

Furthermore, there is a higher turbulent kinetic energy region behind the pier, indicating that a wake vortex is formed. Due to the existence of the negative pressure gradient, the boundary layer separation occurs around the pier. In addition, at this time, the separation surface where the streamlines are concentrated is high velocity and low pressure, while in the place where the streamlines are scattered is the opposite, resulting in the deflection of streamlines and the wake vortex.

It can be clearly seen from Figures 18b and 19b that after installing the ring-wing plate, the turbulent kinetic energy in front of the pier is greatly increased, especially below the plate, reaching 0.058 $m^2s^2$, which is significantly larger than the first condition. This phenomenon can be attributed to the fact that the ring-wing plate, pier, and riverbed form a small semi-enclosed area on this plane, which leads the down-flow to hit the riverbed and rise upwards. In addition, it is blocked by the ring-wing plate and descends again. It may be reasonable that the flow field in this area is very complex due to the horseshoe vortex and the acceleration of the flow. Of note, the location of the high turbulent kinetic energy region at Y = −2 cm is the same as Y = 0. When Figure 19b is examined within this perspective, it can be concluded that there is also a high turbulent kinetic energy region behind the pier at Y = −2 cm, which does not exist for the single pier. It can be seen that the

existence of the ring-wing plate increases the turbulent kinetic energy before and behind the pier, which increases the possibility of scouring the riverbed around the pier.

As shown in Figures 18c and 19c, under the protection of the combined device, the lower turbulent kinetic energy region only generated at the front end of the perforated baffle and the pier near the riverbed. The angled and perforated baffles have the effect of diversion and permeability at the same time, which allows part of the flow to move along the baffle and part of the flow to pass through the baffle. Consequently, only a small part of the flow drops in front of the baffle, resulting in a lower turbulent kinetic energy region. Namely, the baffle permeability increases the efficiency of the combined device. Due to the shielding effect of baffle, pier, and ring-wing plate, the flow field under the plate is very weak and relatively stable, which results in very weak bed scouring around the pier. Moreover, there is almost no apparent turbulent kinetic energy region behind the pier, which is due to the diversion effect of the baffle in front of the pier, thus the flow around the pier is weakened, and the wake vortex is weakened accordingly. Based on the results of the study, it can be asserted that the riverbed around the pier is effectively protected.

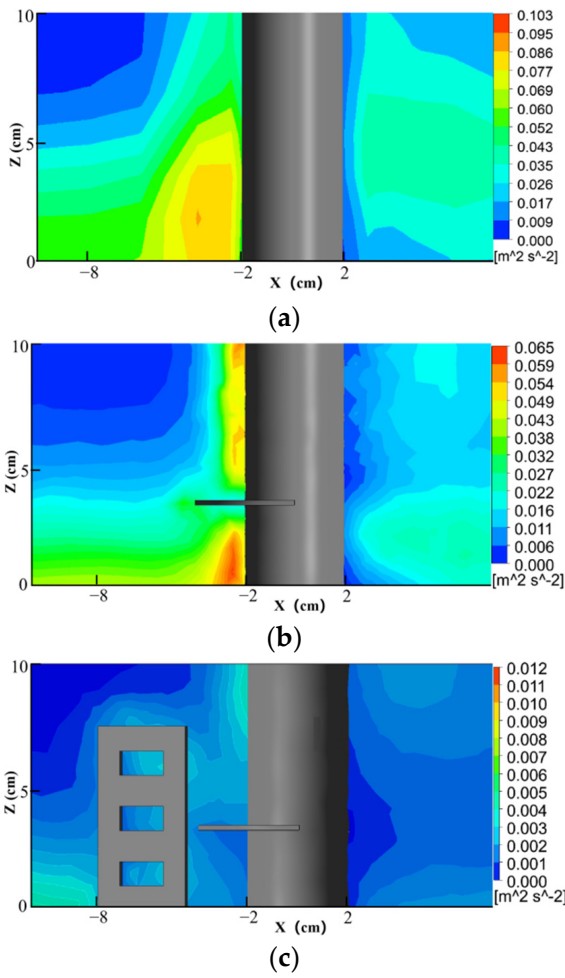

**Figure 18.** Turbulence kinetic energy at Y = 0. (**a**) Single pier; (**b**) Pier with ring-wing plate; (**c**) Pier with combined device.

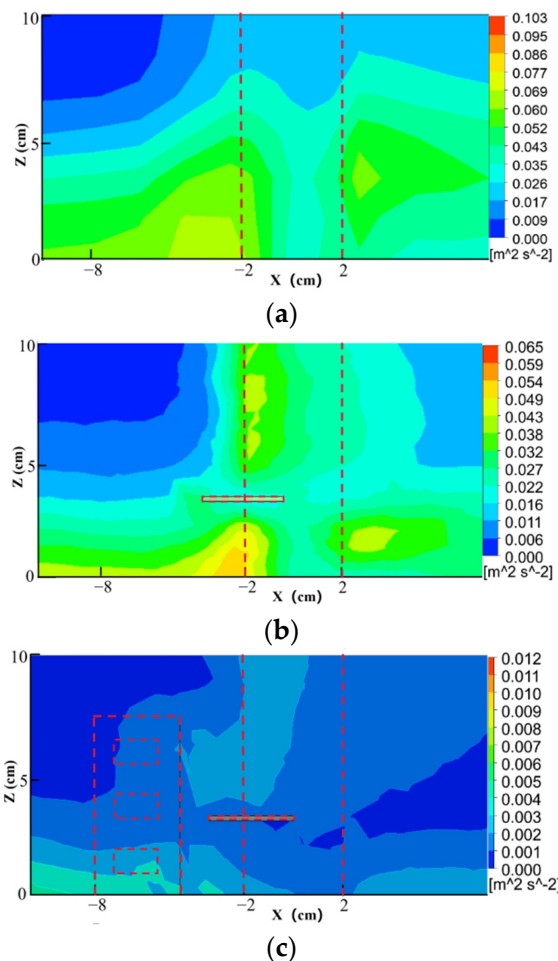

**Figure 19.** Turbulence kinetic energy at Y = −2 cm. (**a**) Single pier; (**b**) Pier with ring-wing plate; (**c**) Pier with combined device.

## 5. Conclusions

In the present study, the combined anti-scour device with the optimal dimensions obtained from the orthogonal experiment was verified by FLUENT. Compared with the single pier and single ring-wing plate, the characteristics of scour hole, velocity vectors, streamlines, and turbulent kinetic energy on different horizontal planes and longitudinal sections are analyzed, and the major conclusions are derived as follows:

- Compared with the single pier and the single ring-wing plate, the depth of the scour hole around the pier under the protection of the combined device is more average, indicating that the riverbed around the pier is well protected together.
- Under the protection of the optimal combined device (S = 20%, L = 2d, H = 1/3h), the maximum scour depth at the front and side of the pier are only 0.5 and 0.4 cm. Compared with the single pier, the corresponding maximum scour depth reduction rates are 84.20% and 78.95%. The anti-scour effect is better than any other group in the orthogonal experiment. In addition, compared with the single ring-wing plate, the anti-scour effect of the combined device is improved by 35.7%.
- The study of the flow field at the horizontal plane shows that under the protection of a single ring-wing plate, the accelerated flow (0.30 m/s) around the pier side and the high turbulent kinetic energy region (0.054 $m^2s^2$) still exist. Due to the diversion of the perforated baffle, the region with high velocity is separated from the pier, and the wake behind the pier is reduced accordingly. Therefore, the turbulent kinetic energy at Z = 1 cm and Z = 5 cm is reduced greatly.

- In the vertical plane, the ring-wing plate effectively reduces the down-flow in front of the pier, but it increases the turbulent kinetic energy. However, after installing the perforated baffle, the turbulent kinetic energy and velocity value in front of and behind the pier is almost 0, which improves the anti-scour effect of the device directly.
- The perforated baffle in the combined device separates the accelerated flow from the bridge pier, reducing the possibility of the sediment on the pier side to be scoured due to excessive bed shear stress. On the basis of water blocking and diversion, the permeability of the perforated baffle effectively reduces the down-flow and turbulent kinetic energy in front of the pier. Therefore, the riverbed in front of the device is also protected, which improves the safety and efficiency of the device.
- In this paper, the formation of scouring around the cylindrical pier and anti-scour mechanism of combined devices are determined. The outcomes from this study may be applied as a valuable data bank for promoting related works on pier anti-scour engineering. The combined device can be applied for both new and existing bridge piers using prefabricated construction technology. If well designed, the perforated baffle and ring-wing plate are cost-effective and can be an appropriate solution in eliminating the local scour. However, the findings of the present study are limited to the flow and soil conditions considered. A more comprehensive study encompassing a wider range of flow conditions, soil, and structure dimensions needs to be undertaken. Furthermore, common specifications and guidelines need to be developed on the use of the combined device before field application.

**Author Contributions:** Funding acquisition, resources, supervision, writing—original draft, Y.W.; conceptualization, methodology, software, investigation, formal analysis, writing—original draft, J.C.; writing—review and editing, Z.W.; investigation, Z.Z.; software, J.Y. All authors have read and agreed to the published version of the manuscript.

**Funding:** This research was funded by the science and technology projects of Zhejiang Provincial Department of transportation in 2019 (Grant No. 2019022).

**Conflicts of Interest:** The authors declare no conflict of interest.

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
