# Peer review of "Numerical Investigation on Local Scour and Flow Field around the Bridge Pier under Protection of Perforated Baffle and Ring-Wing Plate"

_buildings, doi:10.3390/buildings12101544_

Round 1

Reviewer 2 Report

This paper presents the numerical investigation of the local scour and flow field of a series of numerical experiments of the anti-scour device combined with a perforated baffle and ring-wing plate. Some significant conclusions are given and potentially helpful for researchers in the relevant areas. In the reviewer's opinion, the paper should be minor modified before the paper is accepted, and the authors were suggested to clarify the following aspects:

1. This article is based on the experimental data of a reference paper. The authors are suggested to supplement the related experimental photos to show the experimental configuration and equipment set-up situation.

2. How to decide the angle of the perforated baffle in front of the bridge pier? Please add the reason for not including the baffle angle as the optimization parameter. Is this anti-scour device only suitable for circle piers?

3.   In equation 1, the authors should explain whether this definition of error calculation has physical meaning. Figure 5 shows a difference in the flow conditions in front of the bridge pier. How to explain the result?

4.   In table 3, the authors should provide additional information on each parameter's definition and the data source in the table.

5.    Please add the flow velocity distribution profile assumed by the analytical mode. Is there no effect of scour depth variation in the analysis of the simulated flow profile? Is there any consideration of the interaction between soil and water?

6. It is proposed to explore the feasibility or limitation of the proposed method for actual bridge disaster prevention works. The application difficulty seems to be very high. Is there a scaling effect in the analysis?

Round 2

Reviewer 1 Report

The Authors have addressed all the comments suggested by the reviewer. Now, the manuscript can be accepted in the present form

Reviewer 2 Report

The last paragraph of the conclusion should summarize the main contributions of the study and suggestions for future research analysis or experiments.
